# Biological Nanofertilizers to Enhance Growth Potential of Strawberry Seedlings by Boosting Photosynthetic Pigments, Plant Enzymatic Antioxidants, and Nutritional Status

**DOI:** 10.3390/plants12020302

**Published:** 2023-01-09

**Authors:** Said M. El-Bialy, Mohammed E. El-Mahrouk, Taha Elesawy, Alaa El-Dein Omara, Fathy Elbehiry, Hassan El-Ramady, Béni Áron, József Prokisch, Eric C. Brevik, Svein Ø. Solberg

**Affiliations:** 1Soil and Water Department, Faculty of Agriculture, Kafrelsheikh University, Kafr El-Sheikh 33516, Egypt; 2Horticulture Department, Faculty of Agriculture, Kafrelsheikh University, Kafr El-Sheikh 33516, Egypt; 3Agriculture Microbiology Department, Soil, Water and Environment Research Institute (SWERI), Sakha Agricultural Research Station, Agriculture Research Center (ARC), Kafr El-Sheikh 33717, Egypt; 4Department of Basic and Applied Sciences, Higher Institute for Agricultural Cooperation, Cairo 11241, Egypt; 5Institute of Animal Science, Biotechnology and Nature Conservation, Faculty of Agricultural and Food Sciences and Environmental Management, University of Debrecen, 138 Böszörményi Street, 4032 Debrecen, Hungary; 6College of Agricultural, Life, and Physical Sciences, Southern Illinois University, Carbondale, IL 62901, USA; 7Faculty of Applied Ecology, Agricultural Sciences and Biotechnology, Inland Norway University of Applied Sciences, 2401 Elverum, Norway

**Keywords:** catalase, chlorophyll, copper, selenium, peroxidase, photosynthetic pigments, polyphenol oxidase, sustainability

## Abstract

Strawberry production presents special challenges due the plants’ shallow roots. The rooting stage of strawberry is a crucial period in the production of this important crop. Several amendments have been applied to support the growth and production of strawberry, particularly fertilizers, to overcome rooting problems. Therefore, the current investigation was carried out to evaluate the application of biological nanofertilizers in promoting strawberry rooting. The treatments included applying two different nanofertilizers produced biologically, nano-selenium (i.e., 25, 50, 75, and 100 mg L^−1^) and nano-copper (i.e., 50 and 100 mg L^−1^), plus a control (untreated seedlings). The rooting of strawberry seedlings was investigated by measuring the vegetative growth parameters (root weight, seedling weight, seedling length, and number of leaves), plant enzymatic antioxidants (catalase, peroxidase, and polyphenol oxidase activity), and chlorophyll content and its fluorescence and by evaluating the nutritional status (content of nutrients in the fruit and their uptake). The results showed that the applied nanofertilizers improved the growth, photosynthetic pigments, antioxidant content, and nutritional status of the seedlings compared to the control. A high significant increase in nutrient contents reached to more than 14-fold, 6-fold, 5-folf, and 4-fold for Cu, Mn, N, and Se contents, respectively, due to the applied nanofertilizers compared with the control. The result was related to the biological roles of both Se and CuO in activating the many plant enzymes. Comparing the Se with the CuO nanofertilizer, Cu had the strongest effect, which was shown in the higher values in all studied properties. This study showed that nanofertilizers are useful to stimulate strawberry seedling growth and most likely would also be beneficial for other horticultural crops. In general, the applied 100 ppm of biological nano-Se or nano-CuO might achieve the best growth of strawberry seedlings under growth conditions in greenhouses compared to the control. Along with the economic dimension, the ecological dimension of biological nanofertilizers still needs more investigation.

## 1. Introduction

Many environmental risks have occurred, especially the pollution of groundwater and soil, due to the intensive use of traditional chemical/mineral fertilizers [1]. Nanofertilizers are a sustainable approach that can improve crop production and soil quality, as well as reduce many environmental problems [2]. Nanofertilizers can be defined as macro- and micro-nutrients that are present on a nanometer scale and can be taken up by plants in a controlled manner [3,4]. Nanofertilizers can reduce the leaching of nutrients by controlling and/or delaying the release of nutrients and their volatilization into the atmosphere [5]. While physical and chemical methods can be used to produce nanofertilizers, the biological approach has attracted much attention due to the low toxicity and high efficacy of the resulting nutrient sources [6]. Foliar or soil application of bio-nanofertilizers can support crop production under abiotic and/or biotic stress by improving the nutritional status and antioxidant system in treated plants. These benefits have been documented with selenium (Se) [6,7,8,9,10], copper (Cu) [11], and zinc (Zn) [12,13,14,15]. Under natural conditions and human activities, the distribution of NPs in different agroecosystems is considered an important global issue and still needs more extensive investigations in the hydrosphere, including in groundwater, lakes, rivers, and pore water [16], the atmosphere [17], the lithosphere, including mainly soils and sediments [18], and in the biosphere, including mainly plants, micro-organisms, and even humans [19,20].

Nanofertilizers (NFs) are considered an efficient, innovative, and eco-friendly alternative to chemical fertilizers, which should maximize their benefits and minimize their risks [21]. Several studies have confirmed the positive roles of Se bio-nanofertilizers in mitigating environmental stresses because of their high bioavailability, low toxicity, and excellent antioxidant attributes [22]. The role of bio-nano-Se has been investigated for crops such as florist’s daisy (*Chrysanthemum* × *morifolium* (Ramat.) Hemsl.) under high temperature stress [7], cucumber (*Cucumis sativus* L.) under salinity and heat stress [8], pak choi (*Brassica chinensis* L.) under heavy metal stress [10], and rapeseed (*Brassica napus* L.) under salinity stress [9]. There are studies on bio-nano-copper as a guard against plant pathogens such as on absinthium (*Artemisia absinthium* L.) [23], ginger (*Zingiber officinale* Roscoe) [24], and guar (*Cyamopsis tetragonoloba* L.) [25]. However, there are very few studies on the effects of Se and CuO bio-nanofertilizers on other crops such as banana [11] and tomato [26]. The uptake of nanofertilizers through soil or foliar application mainly depends on the characterization of the soil, plant species, and the kind of nanofertilizer [27]. The uptake of nanofertilizers starts through plant cell walls, which are working as the barrier for self-protection (pore size from 5 to 20 nm) via the symplast or apoplast pathway. The size/diameter of the applied nanofertilizer is a vital factor in controlling this uptake, as nanofertilizers can be transported into plant tissues/cells with higher mobility than conventional water-soluble fertilizers [27]. This transport of nanoparticles/nanofertilizers is flexible on both soil application by root entry and foliar entry by leaves [28]. Investigating the uptake, translocation, and accumulation of NPs is a critical issue for their safe application in agriculture. Accumulation of NPs in edible tissues of crops has raised concerns about food safety. The concentrations of NPs in plant tissues generally follow the order of root > shoot > fruit > grains after root exposure [29].

Strawberry (*Fragaria* × *ananassa* (Duchesne ex Weston) Duchesne ex Rozier) is one of the most widely consumed berries worldwide [30]. *Fragaria* is a part of the family *Rosaceae*, which includes several economically important fruit crops such as apple, pear, and peach [31]. Strawberry fruits are very beneficial to human health because of their high ascorbic acid, fiber, plant-derived antioxidants, and micro-nutrient contents [31]. Strawberry plants are characterized as sensitive plants to abiotic/biotic stresses, especially during the seedling stage and based on their shallow roots. The plants are also vulnerable during periods of flowering, fruit setting, and ripening [30]. Strawberry seedling growth can be enhanced by applying amendments such as melatonin under Cd stress [32], auxin application (indole-3-acetic acid; IAA) in saline soils [33], CeO_2_ nanoparticles [34], nutrients such as molybdenum [35], nano-zeolite [36], nano-calcium [37], and other nanofertilizers [38], organic fertilizer after soil fumigation [39], and 5-aminolevulinic acid under salt stress [40].

This is the first reported research, as far we know, concerning the role of biological nano-Cu and nano-Se in promoting the growth of strawberry seedlings. As such, the main objectives of this investigation were to (1) explore the effect of different doses of bio-nano-Se or bio-nano-CuO on the growth of strawberry seedlings; (2) evaluate the effect of the bio-nanofertilizers on enzymatic antioxidants in strawberry seedlings; and (3) investigate the optimal dose of the studied bio-nanofertilizers to achieve the best nutritional status in strawberry seedlings under the conditions studied. The ecotoxicological dimension will be still a crucial issue in nanotechnological studies, especially for nanofertilizers.

## 2. Materials and Methods

### 2.1. Preparing the Bio-Nanofertilizers

Bio-nanoparticles of Se and CuO were produced at the Agricultural Microbiology Laboratory, SWERI, ARC, Giza, Egypt. The Cu and Se were biosynthesized, resulting in nanoparticles that were 87.7 and 41–102 nm, respectively. High-resolution transmission electron microscopy (HR-TEM, Tecnai G20, FEI, Amsterdam, The Netherlands) was used to measure the size of the nanoparticles at the Nanotechnology and Advanced Material Central Laboratory, ARC. The CuO bio-nanofertilizer was prepared using *Bacillus circulans* NCAIM B. 02324 [41], and *Bacillus cereus* TAH was used to prepare the Se bio-nanofertilizer [42].

### 2.2. Applied Treatments

Nano-selenium was applied to the soil in four doses: 0, 25, 50, 75, and 100 mg L^−1^ (treatment codes T2-T5). Nano-copper was applied in two doses: 50 and 100 mg L^−1^ (treatment codes T6 and T7). A control with no CuO (0 mg L^−1^) or Se (treatment code T1) was added (Table 1A,B; Figure 1). Seedlings were planted on 10 April 2021 and the first doses of both nanofertilizers were applied to the soil. After 30 days, a second dose of these nanofertilizers was applied. The plants were grown under a 50% shade net and a fertilizer solution (N-P-K; 19:19:19) was applied once 15 days after culture as a water-soluble fertilizer at a rate of 1 g L^−1^ through the irrigation water (each plastic cup containing a strawberry seedling received 10 mL of fertilizer solution). After 45 days, the experiment was terminated and vegetative parameters were recorded, including survival rate (%), dry weight of seedlings (g), number of leaves per seedling, seedling height (cm), and root length (cm) per seedling.

### 2.3. Growth Medium and Plant Materials

Cold shoots of strawberry (cv. ’Festival’ (Florida Festival (FL 95 41), University of Florida, Gainesville, FL, USA), were kept in a refrigerator for 3 months (January, February, and March 2021) after being imported from the USA. These shoots were leafless and 7–9 cm height. A growing medium was prepared by mixing 1 kg foam + 300 L peatmoss + 30 kg vermiculite. The chemical composition of the peatmoss was pH (4.10), EC (0.189 dS m^−1^), and the nutrient contents (mg L^−1^) were N (323), P (1.8), K (130), Ca (951), Mg (225), Cu (11.6), Mn (28.1), Fe (11.3), and Zn (59.5). A plastic cup (10 cm diameter; volume of each = 412.41 cm^3^) was used as the experimental unit. It was filled with the growing medium which was sterilized using Rizolex fungicide (Sumitomo Chemical Company Ltd., Tokyo, Japan) at a rate of 1.0 g L^−1^ (5 L fungicide for 300 cups). Each plastic cup received 10 mL of fertilizer solution (NPK fertilizer), giving each plant 0.01 g of this compound fertilizer. The fresh water used for irrigation had low salinity (EC 220 dS m^−1^). Calcium carbonate powder was used to adjust the pH of the growth medium to 6 ±1, and then measured with a pH meter (Jenway 3510, Staffordshire, UK). Only one seedling was cultivated in each plastic cup, and each treatment consisted of 5 seedlings or replicates. The study was conducted in the Tahrir region, Badr City (30°08′9.60″ N 31°42′54.00″ E), El Bahira Governorate, Egypt, in a private nursery using an agricultural greenhouse (9 × 40 m).

### 2.4. The Content of Chlorophyll and Its Fluorescence

Young leaves were used to measure the chlorophyll (Chl. a and Chl. b) content with a spectrophotometer (Double beam UV/Visible Spectrophotometer Libra S80PC, England). In brief, the chlorophyll was extracted from leaf tissue by adding 1.0 g of fresh leaf materials to 5 mL N, N-Dimethyl formamide for 48 h in dark conditions at 4 °C. The contents of both chlorophyll types (a and b) were measured through absorbance at 665 and 649 nm. Chlorophyll content was calculated from three replicates for each treatment using the formulae of Lichtenthaler and Welburn [43]. The fluorescence was determined using fresh leaf discs from the abaxial surface by keeping plants in the dark for 30 min before measuring using a portable chlorophyll fluorescence meter (OS30P, Labo Amirica, Fremont, CA, USA). The measured parameters were minimal (F_0_) and maximal fluorescence (F_m_) using light at < 0.1 and > 3500 µmol m^−2^ s^−1^, respectively, on the same leaves. The photochemical efficiency of PSII (F_v_/F_m_) and maximal variable fluorescence (F_v_ = F_m_−F_0_) were calculated according to Dewir et al. [44]. Four randomly chosen expanded young leaves were used to measure 4 single-leaf replicates for each treatment in a standard leaf chamber.

### 2.5. Measurement of Enzymatic Antioxidants

The enzymatic activities were determined using 0.5 g of fully expanded leaves. These leaves were homogenized in a mortar using liquid N with 3 mL of extraction buffer (50 mM TRIS buffer (pH 7.8) containing 1 mM EDTA-Na_2_ and 7.5% polyvinylpyrrolidone)). The homogenate was filtered using 4 layers of cheesecloth under centrifugation at 12,000 rpm for 20 min at 4 °C. The supernatant was re-centrifuged again at 12,000 rpm for 20 min at 4 °C, and a UV-spectrophotometer (160 A—Shimadzu, Kyoto, Japan) was used for measuring total soluble enzyme activity according to Hafez et al. [45]. Polyphenol oxidase activity (EC 1.10.3.1), catalase activity (EC 1.11.1.6), and peroxidase activity (EC 1.11.1.7) were determined using 3 replicates according to the methods described by Malik and Singh [46], Aebi [47], and Hammerschmidt et al. [48], respectively.

### 2.6. Nutritional Status in the Growth Medium and Seedlings

The chemical composition of the growth medium was evaluated before starting the experiment by measuring the soil salinity (electrical conductivity; EC) and pH in 1:1 and 1:5 medium: water ratios using EC (MI 170, Milan, Italy) and pH (Jenway 3510, Staffordshire, UK) meters, respectively. Available potassium (K) and phosphorus (P) were determined using a flame photometer (Jenway PFP7, Staffordshire, UK) and spectrophotometer (GT 80+, Livingston, UK), respectively. An atomic absorption spectrophotometer (Avanta E, GBC, Victoria, Australia) was used to measure the available Se, Cu, manganese (Mn), iron (Fe), and zinc (Zn), as well as nitrogen (N), according to Page et al. [49]. The available concentration of the examined nutrients was extracted using ammonium bicarbonate diethylene tri-amine-penta-acetic acid (AB-DTPA) with 1 M NH_4_HCO_3_ + 0.005 M DTPA solution according to Soltanpour and Schwab [50]. All nutrients, whether digested, AB-DTPA-extracted, or extracted from plant tissues, were measured by atomic absorption spectrometry (AAS). Soil-available P was extracted with ammonium bicarbonate-diethylene triaminepentaacetic (AB-DTPA) and measured calorimetrically by the ascorbic acid method in a + T80 UV-Visible spectrophotometer (PG Instruments, Leicestershire, UK). Seedling samples were oven-dried at 65 °C for 48 h to determine dry weight (DW) and ground in a metal-free mill (IKa-Werke, M 20, Darmstadt, Germany) to obtain a homogenous powder. Seedling dry weight was recorded to calculate nutrient content. Sulfuric acid and hydrogen peroxide were used for wet digestion of the plant samples. Selenium content in seedling samples and the growth medium was measured according to Dernovics et al. [51] using hydride generation atomic fluorescence spectroscopy.

### 2.7. Statistical Design and Analyses

Complete randomized design was implemented using three replicates of each treatment. Each replicate was represented by five seedlings. Data were analyzed using SPSS software (version 20; IBM Corp., Armonk, NY, USA). Mean separations were performed using Duncan’s multiple range testing method and significance was determined at *p* ≤ 0.05.

## 3. Results

### 3.1. Changes in pH and EC in Growth Medium

The effects of applying Se and CuO bio-nanofertilizers were evaluated by monitoring selected vegetative growth parameters. The values of pH and electrical conductivity (EC) of the growing medium before planting the strawberry seedlings were 6.0 and 0.53 dS m^−1^, respectively. After finishing the experiment, growth medium pH and EC were measured again (Table 2). Values of pH increased with increasing nano-Se doses until the 75 mg L^−1^ dose, although this increase was less than one pH unit in all cases. The pH values for the nano-CuO treatments decreased compared to the control as the amount of nano-CuO increased (Table 2). Nano-Se applied at 75 mg L^−1^ led to the highest value in EC (0.479 dS m^−1^), whereas the values of EC significantly decreased with increasing nano-CuO as compared to the control.

### 3.2. Vegetative Growth Parameters

The effects of the applied Se and Cu bio-nanofertilizers were evaluated by monitoring a set of key growth parameters that included seedling height (cm), root length (cm), number of leaves per seedling, and the survival rate (%). In general, these parameters increased with increasing doses of bio-nanofertilizers (i.e., nano-Se and nano-CuO) compared to the control (Table 3). The survival rate (%) expresses the extent to which these nanofertilizers can support the growth and development of the strawberry seedlings. The strawberry seedlings had their highest survival rate at the highest dose of nano-Se (100 mg kg^−1^) combined with either dose of nano-CuO at 96.03, 96.37, and 97.03%, respectively (no statistical difference between survival rates). Using the higher applied doses of the Se nanofertilizer (75 and 100 mg kg^−1^) resulted in the highest root length values (7.43 and 7.77 cm, respectively). These were significantly higher than the control and lowest doses of nano-Se and nano-CuO, but not significantly lower than the highest nano-CuO dose. The results for seedling height were similar, where both doses of nano-CuO and the highest dose of nano-Se resulted in the highest values. Copper nanofertilizer at 100 mg L^−1^ had the highest value of leaves per seedling (5.97).

### 3.3. Photosynthetic Pigments and Their Fluorescence

Chlorophyll pigments and their fluorescence values are given in Figure 2 and Table 3. The measured chlorophyll parameters included Chl. a, Chl. b, and total Chl. The florescence parameters were minimum and maximum fluorescence (F_0_ and F_m_), variable fluorescence (F_v_ = F_m_ − F_0_), (F_V/_F_0_), and (Fv/Fm). There were some remarkable observations: (1) the values of Chl. a for all applied doses of nanofertilizers were higher than Chl. b; (2) a significant increase in chlorophyll parameters (total Chl., Chl. a, and Chl. b) were seen when increasing the applied doses of nanofertilizers; and (3) the Se nanofertilizer increased Chl. values more than the Cu nanofertilizer. Therefore, the highest values of chlorophyll were recorded after applying 100 mg L^−1^ of the Se nanofertilizer. The highest value of total chlorophyll was achieved after applying 100 mg L^−1^ of the CuO nanofertilizer, which was nearly equal to the value found for 25 mg L^−1^ of the CuO nanofertilizer. The CuO nanofertilizer had an obvious impact on the minimum and maximum fluorescence (F_0_ and F_m_) parameters, with significantly higher values (513 and 1713) compared to the control and the Se-nanofertilizer. It is noticeable that the values of (F_V_/F_0_) and (Fv/Fm) decreased with increasing nanofertilizer doses, and the control recorded the highest values of both parameters (Table 4). The rates of increases for Chl. a, b, and total Chl. were about 50% for each parameter.

### 3.4. Enzymatic Antioxidant Activities

There was a significant difference between the two applied nanofertilizers and their doses for each measured enzyme (Table 5). Nano-Se promoted all studied antioxidants as compared to nano-CuO. The highest values (0.41 μM tetra-guaiacol g^−1^ FW min^−1^, 33.66 μM H_2_O_2_ g^−1^ FW min^−1^, and 2.48 μM H_2_O_2_ g^−1^ FW min^−1^ for PPO, CAT, and POX, respectively) for the studied enzymes resulted from applying 100 mg L^−1^ of Se-nanofertilizer. A positive correlation was found between both Se and CuO nanofertilizer doses and enzyme values. The increases in each enzyme were 42, 65, and 51% as compared to the control for CAT, PPO, and POX, respectively.

### 3.5. Nutritional Status of the Growth Medium

The response of seedlings to nanofertilizers depends on the bioavailability of nutrients in the soil or growth medium. There was a significant positive correlation between nanofertilizer doses and available nutrient contents (Cu, Fe, K, N, Mn, P, Se, and Zn) in the growth medium (Figure 3). The growth medium nutrient contents significantly increased with increasing Se nanofertilizer for all studied nutrients. Available nutrient contents in the growth medium significantly decreased with increasing doses of CuO nanofertilizer for N, Mn, and Zn but significantly increased for Cu, K, and Se. There was a non-significant increase in P. Available N in the growing medium increased by 5 times after applying the highest nano-Se fertilizer (100 ppm) dose as compared to the control and doubled after applying 50 ppm of nano-CuO. This increase in Cu content after applying nano-CuO was highly significant, with an increase of more than 14-fold as compared to the control. Every nutrient studied had a significant increase compared to the control after applying nano-Se or nano-CuO.

### 3.6. Nutrient Composition of the Seedlings

The chemical compositions of strawberry seedlings under different doses of nanofertilizers are shown in Figure 4. The impact of nanofertilizers on strawberry seedling chemical composition was significant for all studied nutrients (Cu, Fe, K, Mn, N, P, Se, and Zn). All these nutrients increased with increasing doses of nanofertilizers except N, which decreased when nano-CuO was increased from 50 to 100 mg L^−1^. The highest significant values of seedling nutrient content for P, K, Cu, Mn, and Zn were seen after applying 100 mg L^−1^ nano-CuO, whereas the highest values of N, Fe, and Se resulted from applying the Se nanofertilizer at 100, 100, and 75 mg L^−1^, respectively. The Mn content in seedlings showed the highest increase rate (6-fold), followed by Se (4-fold), P, and Cu (2-fold) as compared to the control. The increasing content of N, K, and Zn in seedlings was significant as compared to the control but did not increase as much as the other studied nutrients. 

### 3.7. Total Uptake of Nutrients by Strawberry Seedlings

The concentration of the studied nutrients in strawberry seedlings was quantified into total uptake using weight of the seedling dry matter (Figure 5). Important findings include the following: (1) all nutrients had a significant correlation with the dose of Se nanofertilizer, while the CuO nanofertilizer had the same trend for all nutrients except Se; (2) the highest values for all nutrients in seedlings were found with the Se nanofertilizer, except for Cu; and (3) the accumulation of nutrients in the seedlings were in the order Fe > N > K > Mn > Zn > P > Cu > Se. Total P uptake by strawberry seedlings was significantly less, by about 30%, in the control compared to nano-Se applications up to 100 ppm, whereas nano-CuO application provided a significant increase of about 10% compared to the control. Iron uptake doubled after applying 75 ppm nano-Se as compared to the control. Iron uptake was about half that with application of nano-CuO. The total Mn uptake was about 4.5 times greater with nano-CuO (100 ppm) application as compared to the control.

## 4. Discussion

Strawberry production depends on agronomic management from planting to harvest. Important production decisions start at the seedling stage and continue through harvest, which may take only 2–3 months [27]. This short period requires a proper fertilization program to produce quality berries but also save macro- and micro-nutrients and meet the physiological and photosynthetic demands of the plant. The shallow root system of strawberry creates a special fertilization program concern. To obtain a more sustainable fertilization, new approaches such as bio-nanofertilizers need to be considered [52]. Thus, the current study focuses on the rooting stage of strawberry seedlings and their response to two biological nanofertilizers, framed by the desire to develop a sustainable agricultural system to produce safe and healthy food.

A strong relationship between nanofertilizers and crop growth has been reported in the literature for banana [11], strawberry [33], maize [53], and rosemary [54]. As far we know, however, there are no published reports concerning the effects of soil-applied CuO or Se nanofertilizers on the rooting of strawberry seedlings. In the current study, the role of two biologically produced nanofertilizers in the growth and development of strawberry seedlings under nursery conditions was investigated. All examined parameters showed significantly increased growth or activity due to both Se and CuO nanofertilizers. Nano-Se gave statistically significant higher values for Chl. A, Chl. B, total Chl., seedling height, the three antioxidant activities: soil N and Se availability; N, Fe, and Se content (mg kg^−1^) in seedlings; and N, P, K, Fe, Mn, Zn, and Se total uptake in seedlings. On the other hand, nano-CuO gave statistically significantly higher values for dry weight of seedlings, F0, Fm, Fv, soil P and Cu availability, P, K, Mn, Zn, and Cu content (mg/kg) in seedlings, and Cu total uptake in seedlings. There are also some measurements where there were no significant differences between Cu and Se. These improvements in seedling growth may be attributed to both Cu and Se as plant essential micro-nutrients that have direct and indirect roles in several plant physiological processes [55], as well as preventing plant damage from environmental stresses and promoting photosynthesis [56,57]. The positive impact of nanofertilizers was confirmed by Shalaby et al. [22], as well as in studies on strawberry [33,35] and rosemary [54].

Photosynthetic pigments and the fluorescence system in plants represent the main factory where they form their food. The efficiency of this system depends on the supply of water, CO_2_, light, nutrients, and indigenous enzymes in plants. Nanofertilizers have a strong relationship with plant photosynthetic pigments, as reported in many studies (e.g., [7,8,53,58]). The CuO nanofertilizer led to higher fluorescence values (Fo, Fm, and Fv) than the Se nanofertilizer or the control. This may be attributed to the role of copper in transporting photosynthetic electrons and Cu ions’ support of the photosynthetic activity of the water splitting system. Copper has the ability to promote the photochemistry of photosystem II, which might result in decreased CO_2_ uptake in plant cells [59]. Nanofertilizers can increase efficiency of the photosynthetic pigments for respiratory systems and plant photosynthesis through their influence on several coenzymes, proteins, vitamins, and purines in plants [60]. These benefits may explain why nanofertilizers (especially the biological ones) minimize nutrient losses and their leaching into groundwater by increasing their bioavailability to plants. There has been increased interest in recent years in using nanofertilizers to promote healthy food, a safe environment, and sustainability [56].

The applied CuO and Se nanofertilizer doses (up to 100 mg L^−1^) were not toxic to strawberry seedlings. This was shown by significant increases in enzymatic antioxidants (i.e., PPO, CAT, and POX) with increasing nanofertilizer doses. Similar research on other crops also showed that nano-CuO or nano-Se did not negatively influence the studied antioxidants in crops, such as in tomato (*Solanum lycopersicum* L.) [61], florist’s daisy [7], rice (*Oryza sativa*) [62], and banana seedlings [11]. Similar results were reported for other nanofertilizers, such as a ZnO nanofertilizer on maize (*Zea mays* L.) [53] and a K nanofertilizer on squash (*Cucurbita maxima* L.) [63].

The high application rates of the studied nanofertilizers promoted enzyme activities in the strawberry seedlings. This may be linked to the presence of Se and Cu micro-nutrients, as both of them are primary components of many plant enzymes and/or their cofactors such as in catalase, polyphenol oxidase, and peroxidase [11]. Selenium can also regulate the activity of antioxidants and their levels in plants. In general, the impact of nano-Se or nano-CuO depends on the applied dose and preparation method. For example, applying 100 mg L^−1^ of nano-Se decreased the chlorophyll content, causing a peroxidation of the chloroplast membrane in tobacco (*Nicotiana tabacum* L.) due to oxidative stress [64]. Different responses were found when biologically manufactured nano-Se was used with banana [11] and in this study. The same trend is valid in the case of CuO nanofertilizer, where applied doses, methods of preparing, and plant species are the main factors controlling this influence. This behavior has been confirmed in many studies, when CuO nanofertilizer up to 400 mg L^−1^ was reported to cause phytotoxicity and genotoxicity in cucumber seedlings [65]. This toxic impact was also noticed when nano-CuO was applied at doses of 500 and 1000 mg L^−1^ to *Brassica rapa* [66] and rice [67], respectively. In contrast, the lower nano-CuO application rates in this study (50 and 100 mg L^−1^) did not show any evidence of toxic impact on the strawberry seedlings.

Chemical Se and CuO nanofertilizers have been tested on several different crops, whereas biological nanofertilizers have recently gained considerable attention as sustainable and non-toxic nutrient sources in the agro-environment [11]. Many crops have been treated with different biological nanofertilizers, such as ZnO nanofertilizer on wheat (*Triticum aestivum* L.) [68], Se nanofertilizer on pak choi [69], CuO nanofertilizer on lettuce (*Lactuca sativa* L.) [58], and Se nanofertilizer on radish (*Raphanus raphanistrum* subsp. *sativus* (L.) Domin) [6]. However, there are no published articles about Se and CuO nanofertilizers on strawberry. Selenium and Cu have the ability to catalyze protein metabolism and regulate the activities of many enzymes, including catalase, ascorbate peroxidase, and glutathione peroxidase, which can eliminate the ROS under oxidative stress in plants [61]. These enzymatic antioxidant promoting roles may be attributed to the Se and CuO nanofertilizers applied in the current study. The main biological functions of nano-Se in plants under stress and suggested mechanisms may include enhancing Se-containing compounds related to human health [70], increasing crop resistance and/or tolerance to biotic/abiotic stresses [71,72], increasing activity of enzymatic/non-enzymatic antioxidants in stressful plants [57], and restricting the uptake and translocation of heavy metals [73,74,75].

The nutritional status of strawberry seedlings in this study was evaluated by measuring the nutrient content in the growth medium and seedlings as well as the total uptake of studied nutrients by the seedlings. The growth medium pH values decreased from 5.42 (for the control) to as low as 5.24 (for 100 mg L^−1^ of nano-CuO) after applying the CuO and Se nanofertilizers. According to Havlin et al. [69], Fe and Mn should increase in plant availability moving from pH 5.4 to 5.2. However, N, P, and K should decrease in plant availability, and Cu and Zn should not change in availability. In the current study, the trends in nutrient availability may not follow long-established expected trends given changes in pH values. Therefore, the positive influence of the CuO and Se nanofertilizers could be due to plant physiological changes rather than nutrient availability due to pH changes. The application of bio-nanofertilizers significantly improved the rooting of strawberry seedlings, which was reflected in more vegetative growth, greater photosynthetic activity, enhanced enzymatic processes, and increased nutrient uptake by the seedlings. The applied nanofertilizers seemed to increase fertilizer efficiency by 50–70%, as well as increase nutrient supply duration in the growth medium for up to 50 days [15]. The particle sizes of nanofertilizers are less than the pore sizes of leaves and roots, which can enhance the uptake efficiency of soil nutrients [76]. Nanofertilizers can support agro-food production by protecting cultivated plants and improving crop productivity [77], for sustainable agriculture [78], precision agriculture [79], and improvement of food quality, safety [20], and nano-farming [80].

The stability of nanoparticles/nanofertilizers is mainly controlled by the environmental conditions, which include rhizosphere and exudates of roots, soil pH, salinity, soil CEC, soil mater content, and soil pollution [81]. After synthetic NPs are formed, they tend to evolve toward a more stable thermodynamic state, through interacting with surrounding molecules or undergoing physicochemical transformations such as corrosion, aggregation, and dissolution [20]. NPs start linking with biomolecules (e.g., proteins, peptides, nucleic acids, lipids, metabolites of cellular activities, and natural organic matter), which allows the adsorbing of NPs on surfaces immediately upon contact with a living system [82]. Expanding attention in the last decade was given to NP interactions within biological environments, as well as NP interactions with ecological components [20]. Therefore, there is an urgent need for additional studies on bio-nanofertilizers and their impacts on agroecosystems, including appropriate application rates.

## 5. Conclusions

Global food production needs to increase to meet the demands of an increasing world population. Appropriate fertilizer management is crucial to increase agricultural food production. The intensive use of traditional mineral fertilizers has led to several environmental problems. Nanofertilizers have shown promise as sustainable slow-release nutrient sources that can increase plant nutrient uptake efficiency. In this study, biological CuO- and Se-nanofertilizers were applied to strawberry seedlings. These nanofertilizers improved seedling growth due to high nutrient bioavailability and nutrient uptake efficiency and promoted plant physiological and biochemical attributes. The nutritional status of the seedlings in the current study may represent a focal point in strawberry seedling production that is well linked to other physiological and biochemical attributes directly and/or indirectly. Copper oxide and Se nanofertilizers can regulate several plant processes, including metabolism of proteins and other biological activities. The nano-CuO and nano-Se may act as cofactors in several enzymes and/or are considered the main components of many plant enzymes. The application of combined nanofertilizers, such as Se and CuO, still needs more investigation at different concentrations with additional crops, and more studies to explore the interactions of these micro-nutrients with the crops’ physiochemical systems. In general, applied nano-CuO at 100 ppm enhanced several studied parameters of strawberry seedlings, such as the uptake of some nutrients (Cu, Mn, K, and Zn), the survival rate of seedlings, and the dry weight of seedlings.

## Figures and Tables

**Figure 1 plants-12-00302-f001:**
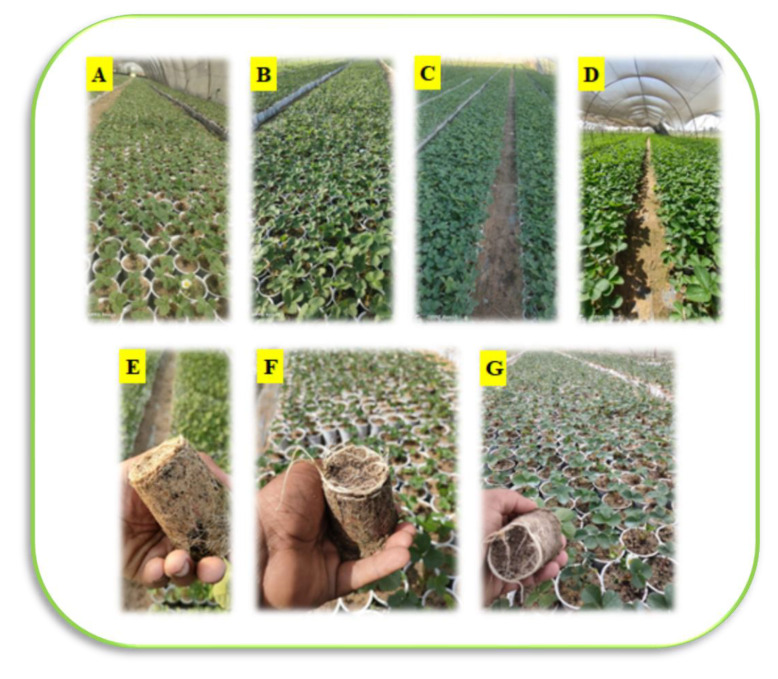
General view of the strawberry seedlings 15, 20, 35 and 45 days after planting (photos (**A**–**D**), respectively). The rooting of these seedlings 20, 25, and 30 days after planting (photos (**E**–**G**), respectively) (photos by El-Baily).

**Figure 2 plants-12-00302-f002:**
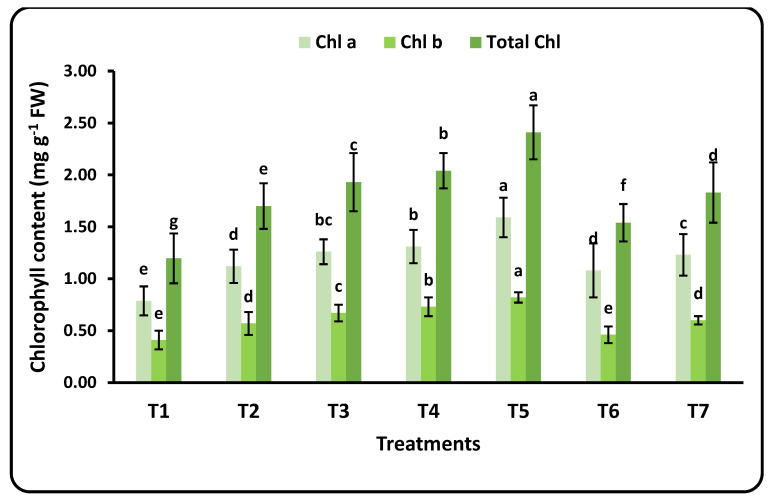
Effects of nanofertilizer dose rate on chlorophyll (Chl. a, Chl. B, and total Chl.) in strawberry leaves. For details about T1 to T7, refer to Table 1. Standard deviation (SD) was calculated from 3 replicates (means ± SD). Means in a given column followed by the same letter are not significantly different at the 5% level.

**Figure 3 plants-12-00302-f003:**
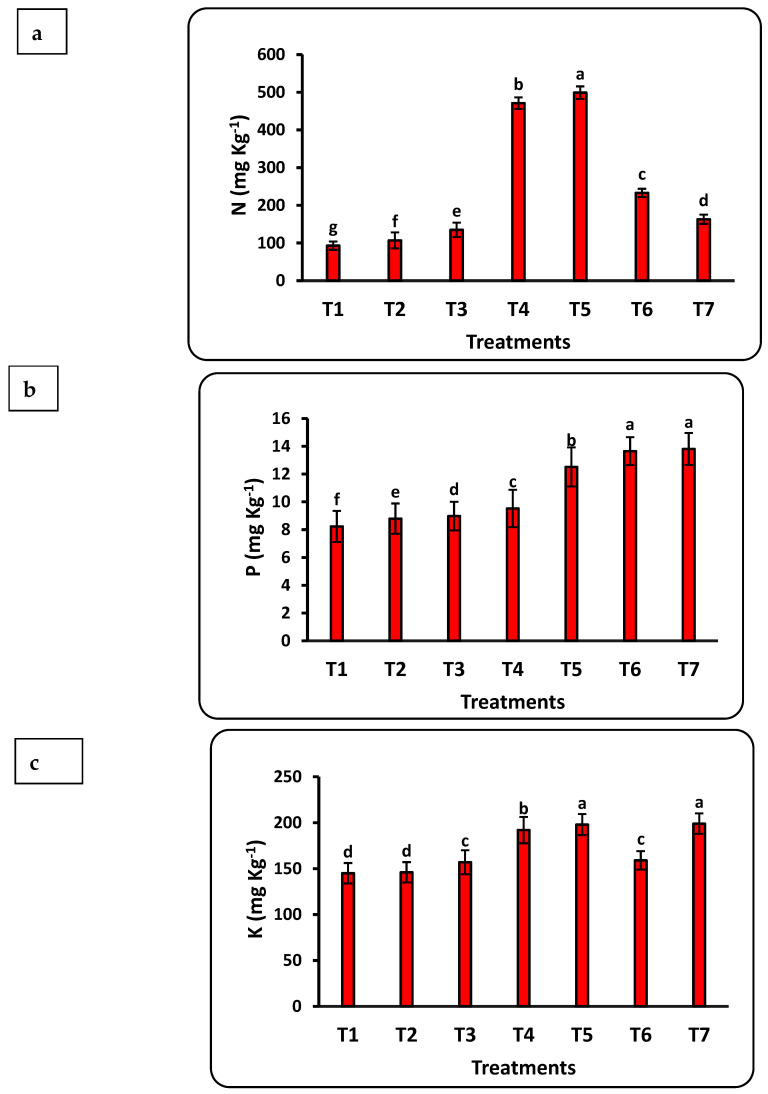
Effects of nanofertilizers on nutrient bioavailability in the growth medium, including N (**a**), P (**b**), K (**c**), Fe (**d**), Mn (**e**), Zn (**f**), Se (**g**), and Cu (**h**). For details about T1 to T7, refer to Table 1. Means in a given column followed by the same letter are not significantly different at the 5% level.

**Figure 4 plants-12-00302-f004:**
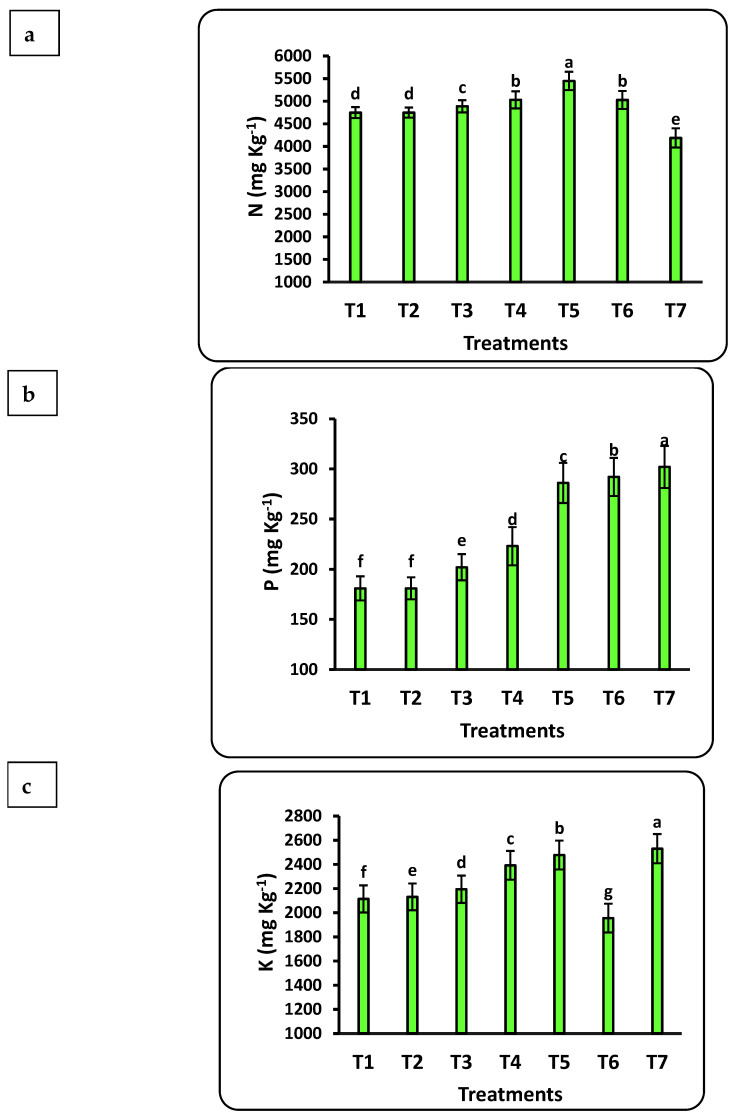
Effects of nanofertilizers on the chemical composition of strawberry seedlings, including N (**a**), P (**b**), K (**c**), Fe (d), Mn (**e**), Zn (**f**), Se (**g**), and Cu (**h**). For details about T1 to T7, refer to Table 1. Means in a given column followed by the same letter are not significantly different at the 5% level.

**Figure 5 plants-12-00302-f005:**
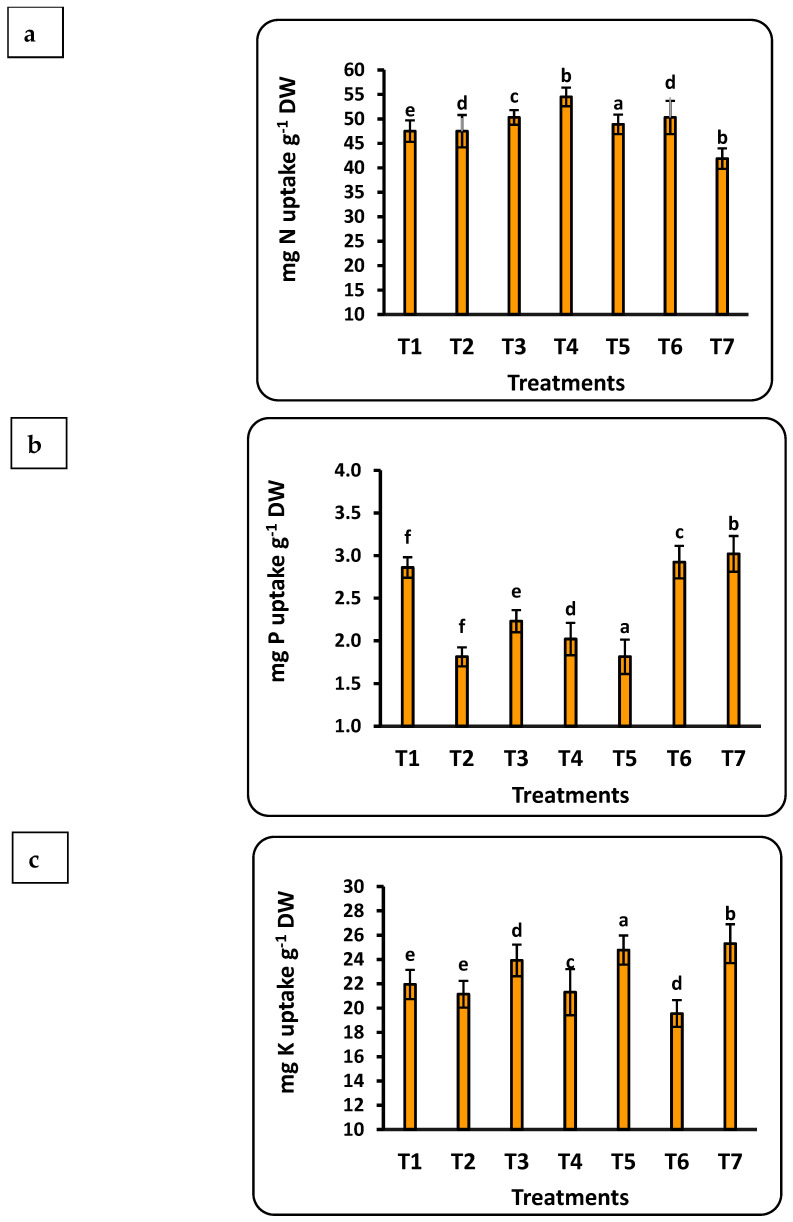
Effects of nanofertilizers on total nutrient uptake in strawberry seedlings, including N (**a**), P (**b**), K (**c**), Fe (**d**), Mn (**e**), Zn (**f**), Se (**g**), and Cu (**h**). For details about T1 to T7, refer to Table 1. Means in a given column followed by the same letter are not significantly different at the 5% level.

**Table 1 plants-12-00302-t001:** (**A**) The fertilizer treatments (NFs = nanofertilizers) applied in the study. (**B**) Schedule of the main agronomic practices performed during the experiment.

**(A)**			
**Code**	**Treatments**	**Applied Doses**	
T1	Control	0 mg L^−1^ Se-NFs; 0 mg L^−1^ CuO-NFs	
T2	Nano-Se	(25 mg L^−1^)	
T3	Nano-Se	(50 mg L^−1^)	
T4	Nano-Se	(75 mg L^−1^)	
T5	Nano-Se	(100 mg L^−1^)	
T6	Nano-CuO	(50 mg L^−1^)	
T7	Nano-CuO	(100 mg L^−1^)	
**(B)**			
**Date after Planting**			**Main Practice**
At planting (time 0)			Application of nanofertilizer treatments
7 days			The first watering was added
14 days			The first dose of mineral fertilizer (NPK) applied (0.01 g NPK fertilizer/seedling)
20 days			Fresh irrigation water applied
25 days			Fresh irrigation water applied
30 days			The second dose of nanofertilizers was applied
34 days			Fresh irrigation water applied
37 days			The second dose of mineral fertilizer (NPK) was applied (0.01 g /seedling)
40 days			Fresh irrigation water applied
43 days			Fresh irrigation water applied
45 days			Measurements of vegetative parameters

**Table 2 plants-12-00302-t002:** The values of both pH and EC and their changes in the growing medium after harvesting strawberry seedlings in each treatment.

Treatments	pH Value of Growth Medium	Δ pH	EC (dS m^−1^) of Growth Medium	Δ EC (dS m^−1^)
Control	5.42 ± 0.12	0.58	0.435 c ± 0.007	0.095
Nano-Se (25 mg L^−1^)	5.32 ± 0.17	0.68	0.446 b ± 0.008	0.084
Nano-Se (50 mg L^−1^)	5.41 ± 0.09	0.59	0.479 a ± 0.007	0.051
Nano-Se (75 mg L^−1^)	5.77 ± 0.11	0.23	0.330 e ± 0.007	0.200
Nano-Se (100 mg L^−1^)	5.52 ± 0.12	0.48	0.329 e ± 0.008	0.201
Nano-CuO (50 mg L^−1^)	5.27 ± 0.13	0.73	0.413 d ± 0.008	0.117
Nano-CuO (100 mg L^−1^)	5.25 ± 0.09	0.75	0.410 d ± 0.008	0.120

Note: Values of pH and EC in the growing medium were 6.0 and 0.53 dS m^−1^ before planting, respectively. Calculated changes: in pH (6.0—treatment value) and in EC (0.53—treatment value). Means in a given column followed by the same letter are not significantly different at the 5% level.

**Table 3 plants-12-00302-t003:** Effects of the different nanofertilizer treatments on dry weight, seedling height, root length, number of leaves per seedling, and survival rate of the strawberry seedlings.

Treatments	Dry Weight of Seedling (g)	Root Length (cm)	Seedling Height (cm)	Leaf No. Per Seedling	Survival Rate (%)
Control	13.37 e ± 0.20	5.97 e ± 0.68	9.53 d ± 1.14	1.66 d ± 0.36	86.19 d ± 3.09
Nano-Se (25 mg L^−1^)	13.58 e ± 0.34	6.67 d ± 0.24	10.23 d ± 1.49	3.30 c ± 1.29	90.70 c ± 1.29
Nano-Se (50 mg L^−1^)	14.12 d ± 0.12	7.13 bc ± 0.27	11.73 c ± 1.56	3.67 bc ± 0.72	91.70 c ± 1.79
Nano-Se (75 mg L^−1^)	14.22 d ± 0.09	7.43 ab ± 0.49	14.00 b ± 1.71	3.97 bc ± 0.15	93.37 bc ± 1.49
Nano-Se (100 mg L^−1^)	14.89 c ± 0.17	7.77 a ± 0.25	16.67 a ± 1.25	4.70 ab ± 0.92	96.03 ab ± 2.08
Nano-CuO (50 mg L^−1^)	15.36 b ± 0.19	6.83 cd ± 0.57	11.86 c ± 1.84	4.93 ab ± 0.39	96.37 a ± 2.52
Nano-CuO (100 mg L^−1^)	16.22 a ± 0.09	7.53 a ± 0.29	13.70 b ± 1.62	5.97 a ± 0.66	97.03 a ± 2.44
L.S.D _0.05_	0.235	0.34	0.94	1.14	293

Means in a given column followed by the same letter are not significantly different at the 5% level. Values are means ± standard deviation (SD) from three replicates (means ± SD).

**Table 4 plants-12-00302-t004:** Effects of different doses of nanofertilizers on fluorescence parameters.

Treatments	F_0_	F_M_	F_V_	F_V/_F_M_	F_V/_F_0_
Control	386 f ± 4.18	1536 f ± 8.81	1174 b ± 4.02	0.76 a ± 0.24	3.04 a ± 0.44
Nano-Se (25 mg L^−1^)	407 e ± 7.55	1560 e ± 3.73	1177 b ± 2.04	0.75 b ± 0.25	2.89 b ± 0.77
Nano-Se (50 mg L^−1^)	433 d ± 7.41	1588 d ± 4.77	1179 b ± 4.13	0.74 c ± 0.98	2.72 c ± 0.14
Nano-Se (75 mg L^−1^)	471 c ± 6.08	1611 c ± 4.62	1164 c ± 2.22	0.72 d ± 0.62	2.47 d ± 0.21
Nano-Se (100 mg L^−1^)	485 b ± 4.53	1619 c ± 5.16	1158 d ± 3.19	0.71 d ± 0.24	2.39 e ± 0.54
Nano-CuO (50 mg L^−1^)	462 c ± 8.72	1663 b ± 8.33	1225 a ± 3.38	0.74 c ± 0.56	2. 65 c ± 0.73
Nano-CuO (100 mg L^−1^)	513 a ± 10.21	1713 a ± 7.15	1224 a ± 2.47	0.71 d ± 0.13	2.39 e ± 0.81
L.S.D _0.05_	12.64	10.60	5.40	0.008	0.07

Means in a given column followed by the same letter are not significantly different at the 5% level. Standard deviation (SD) was calculated for 3 replicates (means ± SD).

**Table 5 plants-12-00302-t005:** Effects of nanofertilizer doses on antioxidant activities including catalase (CAT), polyphenol oxides (PPO), and peroxidase (POX) in strawberry leaves.

Treatments	CAT (μM H_2_O_2_ g^−1^ FW min^−1^)	PPO (μM Tetra—Guaiacol g^−1^ FW min^−1^)	POX (μM H_2_O_2_ g^−1^ FW min^−1^)
Control	19.25 f ± 1.50	0.12 e ± 0.02	1.21 f ± 0.16
Nano-Se (25 mg L^−1^)	22.06 d ± 1.17	0.20 d ± 0.06	1.62 e ± 0.25
Nano-Se (50 mg L^−1^)	26.05 c ± 1.80	0.31 c ± 0.03	2.22 c ± 0.14
Nano-Se (75 mg L^−1^)	29.85 b ± 1.36	0.35 b ± 0.06	2.34 b ± 0.09
Nano-Se (100 mg L^−1^)	33.66 a ± 2.53	0.41 a ± 0.03	2.48 a ± 0.12
Nano-CuO (50 mg L^−1^)	22.96 d ± 1.24	0.27 c ± 0.07	1.93 d ± 0.24
Nano-CuO (100 mg L^−1^)	26.30 c ± 1.54	0.35 b ± 0.08	2.28 bc ± 0.26
L.S.D _0.05_	0.85	0.04	0.08

Means within a given column followed by the same letter are not significantly different at the 5% level. Standard deviation (SD) was calculated for 3 replicates (means ± SD).

## Data Availability

Available with authors at any time.

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
