# Peer review of "Biological Nanofertilizers to Enhance Growth Potential of Strawberry Seedlings by Boosting Photosynthetic Pigments, Plant Enzymatic Antioxidants, and Nutritional Status"

_plants, 2023, doi:10.3390/plants12020302_

Round 1
Reviewer 1 Report
The authors demonstrate positive effects of nanofertilizers on growth and biochemical processes in strawberry. Some of the authors have already demonstrated positive effects of these products in other crops and use strawberry as another case study. Therefore, the
scientific novelty of this study is low, or was at least, not clearly stated.
The authors cite exclusively publications which illustrate positive effects of nanofertilizers on crops although chemical composition of nanofertilizers (namely molecule structures) and
related uptake mechanisms (transporters, synergies, pmf) are unclear. Furthermore, Selenium is considered a beneficial element with some functions in ROS metabolism and
water status, but under which conditions (deficiency of essential elements?) these effects are seen remains rather unclear. Internal transport and use of nanofertilizer molecules is not
clear and would have needed to address more precisely the chemical form of Se, which was
supplied to the crop (elemental form or Se-methionine?). An introduction which is guiding the reader through the scientific state-of-art of nanofertilizer research is missing. This is
particularly relevant, as the guiding questions of this research are not clear. It is not sufficient to illustrate an effect.
The MM section remains rather unclear, which is relevant as nutrient element effects and claims of positive effects require thorough documentation of cultivation and nutrient supply.
Neither pot size nor chemical composition of peatmoss was mentioned. Water supply to crops was not described and supply with nutrients remains unclear. A NPK (water-soluble?)
mineral fertilizer was used, but each pot received 10 ml of a solution which contains 1 g of NPK per liter. It would help to get more meaningful units such as element concentration in
mol/L and total amount of all nutrient elements supplied to plants. Chemical composition of irrigation water was not mentioned. Might that be relevant? How was availability of nutrients
in the substrate measured? Authors should describe the extraction method additionally to the laboratory methods. The authors should try to explain the nutritional status effects of
nanofertilizers on growth medium (Fig. 2), as these effects are remarkable but are not at all explained throughout the text. Figure 4 is with wrong units, if total uptake was to be
presented. The entire discussion is lacking a scientific approach in which observed effects are explained in light of known effects of Se (?) and Co on metabolic activities. Furthermore,
it remains fully speculative to claim that root growth is increased via increased rates of photosynthesis. That holds true under C-source limitations, but likely not in this study with cuttings.
I am sorry to be not more positive in my comments and hope that they help to improve the manuscript.
Author Response
Reviewer 1#
Comments and Suggestions for Authors
The authors demonstrate positive effects of nanofertilizers on growth and biochemical processes in strawberry. Some of the authors have already demonstrated positive effects of these products in other crops and use strawberry as another case study. Therefore, the scientific novelty of this study is low, or was at least, not clearly stated.
Response: Many thanks for your comment!
Yes, we have already confirmed this relationship in previous publications on different crops (below), but as far we know, this is the first work on the impacts of biological nanofertilizers on seedlings production of strawberry! It is also important to note that not all crops respond the same way to the same fertilizer types or application rates. We cannot assume that sugar beet and strawberry, for example, will respond the same way to a 100 mg L-1 application of a given nanofertilizer.
List of our previous published articles on nanofertilizers:
- Abou-Salem, E.; Ahmed, A.R.; Elbagory, M.; Omara, A.E.-D. Efficacy of Biological Copper Oxide Nanoparticles on Controlling Damping-Off Disease and Growth Dynamics of Sugar Beet (Beta vulgaris) Plants. Sustainability 2022, 14, 12871. https://doi.org/10.3390/su141912871
- Ghazi, A.A.; El-Nahrawy, S.; El-Ramady, H.; Ling, W. Biosynthesis of Nano-Selenium and Its Impact on Germination of Wheat under Salt Stress for Sustainable Production. Sustainability 2022, 14, 1784. https://doi.org/10.3390/su14031784
- Seliem, M.K.; Hafez, Y.M.; El-Ramady, H.R. Using Nano - Selenium in Reducing the Negative Effects of High Temperature Stress on Chrysanthemum morifolium Ramat. J. Sus. Agric. Sci. 2020, 46 (3), 47-59. DOI: 10.21608/jsas.2020.23905.1203
- Shalaby, T.A.; Abd-Alkarim, E.; El-Aidy, F.; Hamed, E.; Sharaf-Eldin, M.; Taha, N.; El-Ramady, H.; Bayoumi, Y.; dos Reis, A.R. Nano-selenium, silicon and H2O2 boost growth and productivity of cucumber under combined salinity and heat stress. Ecotoxicol. Environ. Saf., 2021, 212, 111962. https://doi.org/10.1016/j.ecoenv.2021.111962
- Shalaby, T.A.; El-Bialy, S.M.; El-Mahrouk, M.E.; Omara, A.E.-D.; El-Beltagi, H.S.; El-Ramady, H. Acclimatization of In Vitro Banana Seedlings Using Root-Applied Bio-Nanofertilizer of Copper and Selenium. Agronomy 2022b, 12, 539. https://doi.org/10.3390/agronomy12020539
- Saffan, M.M.; Koriem, M.A.; El-Henawy, A.; El-Mahdy, S.; El-Ramady, H.; Elbehiry, F.; Omara, A.E.-D.; Bayoumi, Y.; Badgar, K.; Prokisch, J. Sustainable Production of Tomato Plants (Solanum lycopersicum L.) under Low-Quality Irrigation Water as Affected by Bio-Nanofertilizers of Selenium and Copper. Sustainability 2022, 14, 3236. https://doi.org/10.3390/su14063236
- Zsiros, O.; Nagy, V.; Párducz, Á.; Nagy, G.; Ünnep, R.; El-Ramady, H.; Prokisch, J.; Lisztes-Szabó, Z.; Fári, M.; Csajbók, J.; et al. Effects of selenate and red Se-nanoparticles on the photosynthetic apparatus of Nicotiana tabacum. Photosynth. Res. 2019, 139, 449–460.
- Hassan El-Ramady, Salah E.-D. Faizy, Neama Abdalla, Hussein Taha, Éva Domokos-Szabolcsy, Miklós Fari, Tamer Elsakhawy, Alaa El-Dein Omara, Tarek Shalaby, Yousry Bayoumi, Said Shehata, Christoph-Martin Geilfus, Eric C. Brevik. Selenium and Nano-Selenium Biofortification for Human Health: Opportunities and Challenges. Soil Systems.2020; 4 (3):57.
The authors cite exclusively publications which illustrate positive effects of nanofertilizers on crops although chemical composition of nanofertilizers (namely molecule structures) and related uptake mechanisms (transporters, synergies, pmf) are unclear.
Response: Many thanks for your comment!
You are right, this is a great idea, we will study this in the future. We will follow the transporters for these nanofertilizers and their uptake, this molecular side will be investigated in details by our group as soon as we can. Thanks so much for this advice!
Concerning the mechanism of nanofertilizers uptake, this part was already added to the revise MS:
The uptake of nanofertilizers through soil or foliar application mainly depends on the characterization of soil, plant species and kind of nanofertilizers (Al-Mamun et al. 2021). The uptake of nanofertilizers start through plant cell walls, which are working as the barrier for self-protection (pore size 5 to 20 nm) via the symplast or apoplast pathway. Th size/diameter of applied nanofertilizer is a vital factor control this uptake, which can transport into plant tissues/cells with higher mobility over conventional water-soluble fertilizers (Al-Mamun et al. 2021). This transport of nanoparticles/nanofertilizers is flexible on both soil application by root entry and foliar entry by leaves (Zulfiqar et al. 2019).
Zulfiqar, F.; Navarro, M.; Ashraf, M.; Akram, N.A.; Munné-Bosch, S. Nanofertilizer use for sustainable agriculture: Advantages and limitations. Plant Sci., 2019, 289 (July), 110270. https://doi.org/10.1016/j.plantsci.2019.110270.
Al-Mamun, M.R.; Hasan, M.R.; Ahommed, M.S.; Bacchu, M.S.; Ali, M.R.; Khan, M.Z.H. (2021). Nanofertilizers towards sustainable agriculture and environment. Environ. Technol. Innov., 2021, 23, 101658. doi:10.1016/j.eti.2021.101658
Furthermore, Selenium is considered a beneficial element with some functions in ROS metabolism and water status, but under which conditions (deficiency of essential elements?) these effects are seen remains rather unclear.
Response: Many thanks for your comment!
Based on the published information about the Se and its role on plant growth under stress, Se has dual effects on the uptake of micro-nutrients in plants (including toxicity and deficiency). The protective roles of Se in alleviating element toxicity/deficiency by enhancing plant tolerance system against generation of ROS, improving antioxidant capacity, and pathogens resistance were confirmed as reported by many researchers such as Gui et al. (2022), and Kang et al. (2022). Other articles recently published on this role of Se in cultivated plants under stress including mineral and nanoform of Se include:
Hossain, M.A.; Ahammed, G.J.; Kolbert, Z.; El-Ramady, H.; Islam, T.; Schiavon, M. Selenium and Nano-Selenium in Environmental Stress Management and Crop Quality Improvement. Sustainable Plant Nutrition in a Changing World Book Series, Springer Nature Switzerland, Cham, Switzerlan, 2022
Gui, Jia-Ying, Shen Rao, Xinru Huang, Xiaomeng Liu, Shuiyuan Cheng, Feng Xu (2022). Interaction between selenium and essential micronutrient elements in plants: A systematic review. Science of The Total Environment, 853, 158673. https://doi.org/10.1016/j.scitotenv.2022.158673.
Kang, L, Yangliu Wu, Jingbang Zhang, Quanshun An, Chunran Zhou, Dong Li, Canping Pan (2022). Nano-selenium enhances the antioxidant capacity, organic acids and cucurbitacin B in melon (Cucumis melo L.) plants. Ecotoxicology and Environmental Safety, 241, 113777. https://doi.org/10.1016/j.ecoenv.2022.113777.
El-Ramady, H, Tamer El-Sakhawy, Alaa El-Dein Omara, József Prokisch, and Eric C. Brevik (2022). Selenium and Nano-Selenium for Plant Nutrition and Crop Quality. In: M. A. Hossain et al. (eds.), Selenium and Nano-Selenium in Environmental Stress Management and Crop Quality Improvement, Sustainable Plant Nutrition in a Changing World, https://doi.org/10.1007/978-3-031-07063-1_4, pp: 55-78.
This also was confirmed for human diseases like:
Tanmoy Rana (2022). Emerging Nano-selenium: An insight to Its Current Status and Potentials in ROS-Induced Cancer Prevention and Therapy. Handbook of Oxidative Stress in Cancer: Therapeutic Aspects
More mechanisms were published in El-Ramady et al. (2022) in our chapter (below Table):
Table 1: The main biological functions of selenium and nano-Se in plants and suggested mechanisms (source: El-Ramady et al. 2022)
|
Biological functions of Se and nano-Se |
Suggested mechanism or clarification |
References |
|
(1) increases crop growth and yield |
Applied Se or nano-Se in proper dose enhances plant growth and yield |
Neysanian et al. (2020); Li et al. (2021a) |
|
(2) promotes crop ripening, senescence and shelf-life |
Increasing Se content in fruits lead to reduce fruit-softening rates, thus increasing the shelf-life of fruit products |
Wen (2021) |
|
(3) enhances Se-containing compounds related to human health |
These compounds may include the 30 known selenoproteins (e.g., selenocysteine) and other organic-Se like seleno-amino acids, seleno-proteins, Se-polysaccharides, etc. |
El-Ramady et al. (2020); Chen et al. (2021); Groth et al. (2020, 2021) |
|
(4) increases crop resistance and/or tolerance to abiotic and biotic stresses |
By regulating the antioxidant system via stimulating photosynthesis; repairing damaged cell structures and their functions; and rebalancing of essential nutrients in plant tissues |
Feng et al. (2021b); Hossain and Islam (2021) |
|
(5) enhances the photosynthesis process (photosynthetic pigments including chlorophyll and carotenoids) and its rate |
Regulates photosynthesis system; Se impacts on PSII and their electron transfer processes; Bio-Se-NPs increases content of photosynthetic pigments because Bio-Se-NPs surface is surrounded with active phytochemicals |
Yin et al. (2019); Hernández-Hernández et al. (2019); Amirabad et al. (2020); Borbély et al. (2021) |
|
(6) increasing activity of enzymatic antioxidants like superoxide dismutase (SOD), catalase (CAT), and peroxidase (APX) |
Selenium can increase plant tolerance to oxidative stress because it is a main component of many enzymatic antioxidants like glutathione peroxidase and SOD, which reduce lipids content |
Rady et al. (2020); Hasanuzzaman et al. (2020); Nawaz et al. (2021) |
|
(7) increases accumulation of non-enzymatic antioxidants e.g., ascorbic acid, flavonoids, proline and tocopherol |
accumulation of osmo-protectants, and secondary metabolites, which act as a scavenging system and promote cell detoxification |
Sabatino et al. (2021); Lanza and Reis (2021); Mateus et al. (2021) |
|
(8) reduces the toxicity of metals/metalloids like Cd, Hg, Pb, and Se itself |
By regulating gene expression, sequestering HMs in the root cell walls and organelles, and reducing HM transfer from roots to the shoots |
Huang et al. (2021); Riaz et al. (2021) |
|
(9) metal detoxification in soil–plant systems |
May reduce HM bioavailability in soil by forming HM-Se-containing complexes in the roots and/or rhizosphere; reducing the uptake of HMs by plant roots and their translocation |
Tran et al. (2018, 2021); Yang et al. (2021) |
|
(10) restricts the uptake and translocation of heavy metals (As, Cd, Cu, Mn, Zn, etc.) in plants |
Through changing root hair morphology by regulating hormones and increasing thickness of cell wall by enhancing contents of cell wall components (e.g., lignin, pectin, hemicelluloses) Se regulates accumulation of HMs in cell wall |
Feng et al. (2021a); Riaz et al. (2021); Wang et al. (2021a) |
|
(11) reduces accumulation and toxicity of heavy metals (As, Cr, Cd, Hg, Pb, etc.) in plants |
Reducing the bioavailability of HMs in soils and then uptake by plants; Se impacts on uptake/the sequestration of HMs; Se-combines with HMs and then sequesters them in plant cells |
Ding et al. (2020); Hussain et al. (2020); Feng et al. (2021b) |
Internal transport and use of nanofertilizer molecules is not clear and would have needed to address more precisely the chemical form of Se, which was supplied to the crop (elemental form or Se-methionine?).
Response: Many thanks for your comment!
In general, the uptake Se-NPs or selenite was rapidly assimilated to organic forms, with Se-Met being the most predominant species in both shoots and roots of the rice plants. However, following selenate treatment, Se(VI) remained as the most predominant species, and only a small amount of it was converted to organic forms (Wang et al. 2020).
First of all, in our study we used nano elemental Se not the organic form of Se. Selenium occurs in multiple valence states (i.e., −2, 0, +2, +4, and + 6), which include the metal selenide (Se−, Se2−), elemental Se or nano-Se (Se0), thio-selenate (SSeO3 2−), selenite (SeO3 2−), and selenate (SeO4 2). Red elemental nano-Se spheres in water may produce H2Se and H2SeO3 in small amounts as shown in Eq. 1.1:
3Se 3+ ↔ HO2 2 2HSe + HS2 3 eO
H2Se and H2SeO3 are formed in solution. When the solution dries, the H2Se and H2SeO3 react with each other and elemental selenium is precipitated, forming crystals (El-Ramady et al. 2022). It is well documented that the transformation of Se and Nano-Se in Soils mainly controlled by the microbial activity along with roots exudates, and soil properties.
Nano-Se has a greater potential than inorganic selenium (like Se-methionine) in preventing Se-deficiency diseases due to its higher safety (Chen et al. 2022).
Sources:
El-Ramady, H, Alaa El-Dein Omara, Tamer El-Sakhawy, József Prokisch, and Eric C. Brevik
(2022). Sources of Selenium and Nano-Selenium in Soils and Plants. In: M. A. Hossain et al. (eds.), Selenium and Nano-Selenium in Environmental Stress Management and Crop Quality Improvement, Sustainable Plant Nutrition in a Changing World, https://doi.org/10.1007/978-3-031-07063-1_1, pp: 1-24
Chen J, Feng T, Wang B, He R, Xu Y, Gao P, Zhang Z-H, Zhang L, Fu J, Liu Z and Gao X (2022) Enhancing organic selenium content and antioxidant activities of soy sauce using nano-selenium during soybean soaking. Front. Nutr. 9:970206. doi: 10.3389/fnut.2022.970206
Wang, K., Wang, Y., Li, K. et al. Uptake, translocation and biotransformation of selenium nanoparticles in rice seedlings (Oryza sativa L.). J Nanobiotechnol 18, 103 (2020). https://doi.org/10.1186/s12951-020-00659-6
An introduction which is guiding the reader through the scientific state-of-art of nanofertilizer research is missing. This is particularly relevant, as the guiding questions of this research are not clear. It is not sufficient to illustrate an effect.
Response: Many thanks for your comment!
We added more in the introduction section to be fit, thanks for your comment!
we provided 2 paragraphs about nanofertilizers, which included the following issues:
- The meaning of these nanofertilizers, why these fertilizers are sustainable especially the biological ones, The uptake of nanofertilizers,
- Role of these fertilizers in reducing environmental pollution compared to chemical fertilizers
- A comparison between chemical and nanofertilizers properties,
- The crucial role of these fertilizers in promoting crop production even under normal and stressful conditions
We have more publications on nanofertilizers as we mentioned above in original research (above) and review articles (below):
Shalaby, T.A.; Bayoumi, Y.; Eid, Y.; Elbasiouny, H.; Elbehiry, F.; Prokisch, J.; El-Ramady, H.; Ling, W. Can Nanofertilizers Mitigate Multiple Environmental Stresses for Higher Crop Productivity?. Sustainability 2022, 14, 3480. https:// doi.org/10.3390/su14063480
El-Ghamry ·et al. (2018). Nanofertilizers vs. Biofertilizers: New Insights. https://jenvbs.journals.ekb.eg
The MM section remains rather unclear, which is relevant as nutrient element effects and claims of positive effects require thorough documentation of cultivation and nutrient supply.
Response: Many thanks for your comment!
We added tables of treatments and a figure including the agronomic practices during this work in an effort to address this concern!
Neither pot size nor chemical composition of peatmoss was mentioned.
Response: Many thanks for your comment!
We mentioned in the MS that “A certain growing medium composed by mixing 1 kg foam + 300 L peatmoss + 30 kg vermiculite.”
Peatmoss analysis was as follows:
|
Parameter |
N (mg/L) |
P (mg/L) |
K (mg/L) |
Mg (mg/L) |
Ca (mg/L) |
Fe (mg/L) |
Mn (mg/L) |
Zn (mg/L) |
Cu (mg/L) |
Na (mg/L) |
pH |
EC (dS/m) |
|
Value |
323.7 |
1.8 |
130 |
225 |
951 |
11.38 |
28.12 |
59.5 |
11.6 |
158.8 |
4.1 |
0.189 |
The volume or size of pot was 412.41 cm3. Information on the peat moss chemical composition and pot size has been added to the manuscript.
Water supply to crops was not described and supply with nutrients remains unclear.
Response: Many thanks for your comment!
All irrigation and fertilization schedule and method of apply were mentioned in the added figure as follows:
- At planting (0 time) Nano-treatments were applied
- After one week of planting, the first watering was added
- After 2 weeks from planting, the first dose mineral fertilizers (NPK), each plant receives 0.01 mg of compound NPK fertilizer
- After 20 days from planting, fresh irrigation water was added
- After 25 days from planting, fresh irrigation water was added
- After 30 days from planting, the second dose of nanofertilizers was added
- After 34 days from planting, fresh irrigation water was added
- After 37 days from planting, the second dose of nanofertilizers was applied
- After 40 days from planting, fresh irrigation water was added
- After 43 days from planting, fresh irrigation water was added
- After 45 days from planting, data were taken
- The irrigation was applied using watering cans of 10 liter
A NPK (water-soluble?) mineral fertilizer was used, but each pot received 10 ml of a solution which contains 1 g of NPK per liter. It would help to get more meaningful units such as element concentration in mol/L and total amount of all nutrient elements supplied to plants.
Response: Many thanks for your comment!
Simply, each plastic cup received 10 ml from fertilizer solution (the compound fertilizer of NPK), meaning each plant received 0.01 g of this compound fertilizer. This has been added to the manuscript.
Chemical composition of irrigation water was not mentioned. Might that be relevant?
Response: Many thanks for your comment!
The main chemical parameter of irrigation water was the salinity (220 ppm). So, this water was fresh and should not have caused any stress on the seedlings, as we noticed during the study. The salinity information has been added to the manuscript.
How was availability of nutrients in the substrate measured? Authors should describe the extraction method additionally to the laboratory methods.
Response: Many thanks for your comment!
The physicochemical characterization of the soil samples was conducted according to Sparks et al. (1996). The available concentration of the examined nutrients was extracted using ammonium bicarbonate diethylene tri-amine-penta-acetic acid (AB-DTPA) with 1 M NH4HCO3 + 0.005 M DTPA solution according to Soltanpour and Schwab (1977). All nutrients, whether digested, AB-DTPA-extracted, or extracted from plant tissues, were measured by atomic absorption spectrometry AAS (GBC Avanta E, Victoria, Australia). Soil-available phosphorus (P) was extracted by ammonium bicarbonate-diethylene triaminepentaacetic (AB-DTPA) and measured calorimetrically by the ascorbic acid method in a + T80 UV-Visible spectrophotometer (PG Instruments, UK). For plant samples the wet digestion was used using sulfuric acid and H2O2. This information has been added to the manuscript.
Sparks et al. (1996) or Page et al. (1982).
Sparks, D.L. (1996) Methods of Soil Analysis Part 3: Chemical Methods. Soil Science Society of America, American Society of Agronomy, Madison.
Soltanpour, P. N., & Schwab, A. P. (1977). A new soil test for simultaneous extraction of macro- and micro-nutrients in alkaline soils. Communications in Soil Science and Plant Analysis, 8, 195–207.
The authors should try to explain the nutritional status effects of nanofertilizers on growth medium (Fig. 2), as these effects are remarkable but are not at all explained throughout the text.
Response: Many thanks for your comment! Added more explanations, thanks!
Figure 4 is with wrong units, if total uptake was to be presented.
Response: Many thanks for your comment!
The unit of nutrient uptake was correct, please (mg of the nutrient per g dry weight plant material). It was converted from mg per kg plant material dry weight!
The entire discussion is lacking a scientific approach in which observed effects are explained in light of known effects of Se (?) and Co on metabolic activities. Furthermore, it remains fully speculative to claim that root growth is increased via increased rates of photosynthesis. That holds true under C-source limitations, but likely not in this study with cuttings.
Response: Many thanks for your comment!
We added more improvements, thanks!
Both Se and Cu are important in several plant enzymes, which control several biological plant activities and metabolism. Photosynthesis depends on the bioavailability many nutrients like selenium and copper, which indirectly and/or directly can enhance (at lower or suitable dose depending on plant species) or inhibit (at high applied dose) this photosynthesis.
For sure, our study is one of many studies that are needed to investigate the potential of biological nanofertilizers. Sure, several open questions remain concerning this vital issue. But we cannot address all of them in a single paper.
I am sorry to be not more positive in my comments and hope that they help to improve the manuscript.
Response: Many thanks for your comments! Definitely these comments already improved the MS, thanks!

Reviewer 2 Report
The authors presented their work to the journal ‘Plants’. While the topic is interesting, the manuscript needs major revision prior to publication.
Why these conditions for experiments were used during the work? It would be necessary to explain why these nanoparticles concentrations were used in measurements. What was the rationale for this? Are they elemental or oxide particles? They are not of the same size. How can they be compared in the same experiment? Why not the same doses were applied? What about the stability of nanoparticles?
Did the authors examine upward transport from roots to shoots? How did the pH value change during the experiment?
The English is not completely satisfactory; it needs a thorough revision.
Quantification of Cu and Se in plant tissues, the release of Cu and Se from nanofertilizers should be confirmed by measurements, like ICP-MS.
Where were these particles located in plants? It will be nice to see some microscopic images of this as well.
Author Response
Reviewer 2#
Comments and Suggestions for Authors
The authors presented their work to the journal ‘Plants’. While the topic is interesting, the manuscript needs major revision prior to publication.
Response: Many thanks for your comments! We followed your comments to improve the revised MS, thanks again!
Why these conditions for experiments were used during the work? It would be necessary to explain why these nanoparticles concentrations were used in measurements. What was the rationale for this? Are they elemental or oxide particles? They are not of the same size. How can they be compared in the same experiment? Why not the same doses were applied? What about the stability of nanoparticles?
Response: Many thanks for your comments!
First of all, these nanoparticles were prepared using biological methods NOT chemical ones to minimize their toxicity to the growing media or the agro-environment in general. Biological preparation can lead to less uniform NP size.
Secondly, we do not currently know what an appropriate application rate of these nanofertilizers is for strawberry. Some other crops have shown toxicity affects at higher doses (e.g., 500 or 1000 mg L-1), therefore, we tried several lower doses to see what the responses were to them.
Copper was in oxide form (CuO), and Se was elemental selenium. They are not of the same size because producing NP by microbes depends on the method of production, microbe species, and many other factors. In our study microbe species is the dominant factor controlling the size of the NPs.
We can compare these nanoparticles even though they have different sizes because each one is a factor or treatment. The applied dose depends on the cultivated plants and methods of application including soil, foliar or in vitro, or seed priming, etc. and this was confirmed in our previous published articles such as:
- Abou-Salem, E.; Ahmed, A.R.; Elbagory, M.; Omara, A.E.-D. Efficacy of Biological Copper Oxide Nanoparticles on Controlling Damping-Off Disease and Growth Dynamics of Sugar Beet (Beta vulgaris) Plants. Sustainability 2022, 14, 12871. https://doi.org/10.3390/su141912871
- Ghazi, A.A.; El-Nahrawy, S.; El-Ramady, H.; Ling, W. Biosynthesis of Nano-Selenium and Its Impact on Germination of Wheat under Salt Stress for Sustainable Production. Sustainability 2022, 14, 1784. https://doi.org/10.3390/su14031784
- Seliem, M.K.; Hafez, Y.M.; El-Ramady, H.R. Using Nano - Selenium in Reducing the Negative Effects of High Temperature Stress on Chrysanthemum morifolium Ramat. J. Sus. Agric. Sci. 2020, 46 (3), 47-59. DOI: 10.21608/jsas.2020.23905.1203
- Shalaby, T.A.; Abd-Alkarim, E.; El-Aidy, F.; Hamed, E.; Sharaf-Eldin, M.; Taha, N.; El-Ramady, H.; Bayoumi, Y.; dos Reis, A.R. Nano-selenium, silicon and H2O2 boost growth and productivity of cucumber under combined salinity and heat stress. Ecotoxicol. Environ. Saf., 2021, 212, 111962. https://doi.org/10.1016/j.ecoenv.2021.111962
- Shalaby, T.A.; El-Bialy, S.M.; El-Mahrouk, M.E.; Omara, A.E.-D.; El-Beltagi, H.S.; El-Ramady, H. Acclimatization of In Vitro Banana Seedlings Using Root-Applied Bio-Nanofertilizer of Copper and Selenium. Agronomy 2022b, 12, 539. https://doi.org/10.3390/agronomy12020539
- Saffan, M.M.; Koriem, M.A.; El-Henawy, A.; El-Mahdy, S.; El-Ramady, H.; Elbehiry, F.; Omara, A.E.-D.; Bayoumi, Y.; Badgar, K.; Prokisch, J. Sustainable Production of Tomato Plants (Solanum lycopersicum L.) under Low-Quality Irrigation Water as Affected by Bio-Nanofertilizers of Selenium and Copper. Sustainability 2022, 14, 3236. https://doi.org/10.3390/su14063236
- Zsiros, O.; Nagy, V.; Párducz, Á.; Nagy, G.; Ünnep, R.; El-Ramady, H.; Prokisch, J.; Lisztes-Szabó, Z.; Fári, M.; Csajbók, J.; et al. Effects of selenate and red Se-nanoparticles on the photosynthetic apparatus of Nicotiana tabacum. Photosynth. Res. 2019, 139, 449–460.
- Hassan El-Ramady, Salah E.-D. Faizy, Neama Abdalla, Hussein Taha, Éva Domokos-Szabolcsy, Miklós Fari, Tamer Elsakhawy, Alaa El-Dein Omara, Tarek Shalaby, Yousry Bayoumi, Said Shehata, Christoph-Martin Geilfus, Eric C. Brevik. Selenium and Nano-Selenium Biofortification for Human Health: Opportunities and Challenges. Soil Systems.2020; 4 (3):57.
Obviously the results are only known to be valid for the size ranges and doses used, but all of this is reported.
Why were the same doses not applied?
As you can see in the list of published articles, applied doses mainly depend on the plant species and the method of applying (soil or foliar or seed priming, or in vitro…). We found in our previous study that applied doses of nano-Se ranged from 25 to 200 ppm, where nano-CuO was higher and ranged from 50 to 200 ppm or more.
What about the stability of nanoparticles?
The stability of nanoparticles is mainly controlled by the environmental conditions, which include rhizosphere and exudates of roots, soil pH, and salinity, soil CEC, soil mater content, etc.
Did the authors examine upward transport from roots to shoots? How did the pH value change during the experiment?
Response: Many thanks for your comments!
We did not examine upward transport from roots to shoots, but this will be our next work on the nanofertilizers. We will study this upward transport from roots to shoots using different cultivated plants, thanks for your advice!
How did the pH value change during the experiment? This change may back to the exudates of plant roots during the growth beside the soil microbial activities, which may lead to decomposition of soil organic matter, and one more reason like nano Se, which already prepared using acidic solution (HCl) not water, and this may have impact on this change!
The English is not completely satisfactory; it needs a thorough revision.
Response: Many thanks for your comments!
Our co-author Prof. Eric Brevik (from the USA) is a native English speaker and has read the manuscript carefully again for the English.
Quantification of Cu and Se in plant tissues, the release of Cu and Se from nanofertilizers should be confirmed by measurements, like ICP-MS.
Response: Many thanks for your comments!
Yes, this device is great for measuring both Se and Cu, when this is available!
We mentioned in the MS Line-no. 162 that “An atomic absorption spectrophotometer (Avanta E, GBC, Victoria, Australia) was used to measure the available Se, Cu,…..”
Where were these particles located in plants? It will be nice to see some microscopic images of this as well.
Response: Many thanks for your comments! This is great advice but we did not do this because we do not have the facilities needed. We hope we can obtain the necessary equipment and can carry out this very good idea in the future!
Concerning the transport of nanoparticles (NPs) in plants, it depends on many factors such as method of application (root or soil or foliar application), kind of nanoparticles and its size or diameter, etc.
In general, exposure of NPs to plants occurs via foliar or root exposure. The root application of these NPs enables their entrance into plant tissues by crossing root epidermal cell membranes through the apoplastic or symplastic pathway to reach the vascular tissue. For foliar application, plant leaves are the main pathway for NPs-uptake and translocation through the cuticle, which serves as the first defence barrier, and lead to their uptake, translocation and accumulation in plant tissues.
Investigating the uptake, translocation and accumulation of NPs is a critical issue for its safe application in agriculture. Accumulation of NPs in edible tissues of crops has raised concerns about food safety. The concentrations of NPs in plant tissues generally follow the order of root > shoot > fruit > grains after root exposure.
Source:
Ved Prakash, Jose Peralta-Videa, Durgesh Kumar Tripathi, Xingmao Ma, Shivesh Sharma, (2021). Recent insights into the impact, fate and transport of cerium oxide nanoparticles in the plant-soil continuum. Ecotoxicology and Environmental Safety, 112403. https://doi.org/10.1016/j.ecoenv.2021.112403.

Reviewer 3 Report
Experiments were well performed and the theme is moderately novel and very interesting, but some minor improvements are needed. Overall, the manuscript will meet the publishing standard of the journal after minor revisions.
Title: The existing title does not accurately summarize the content of this study, and 'Enhance Rooting' is not comprehensive enough to cover the content, so it is suggested to change it to 'Enhance growth potential'.
Figure 2-4: It is recommended to number the bar charts in the combination chart, such as A, B, C... or a, b, c....
The position format of the horizontal legend 'Treatments' of the bar graph is inconsistent.
Discussion: Mentioned in 'Introduction': “the main objectives of this investigation were to: (3) investigate the optimal dose of the studied bio-nanofertilizers for the best nutritional status in strawberry seedlings under the conditions studied”; And with seven treatments (T1-T7) included in the experimental design, It is recommended that a comprehensive data analysis be conducted to conclude which treatment is more beneficial to the growth of strawberries and reflected in the abstract and conclusion.
Conclusions: It is better to avoid any general discussion in the conclusion part; please try to focus on the principal findings. It is suggested that the general discussion could be placed in 'Discussion'.
Author Response
Reviewer 3#
Comments and Suggestions for Authors
Experiments were well performed and the theme is moderately novel and very interesting, but some minor improvements are needed. Overall, the manuscript will meet the publishing standard of the journal after minor revisions.
Response: Many thanks for your comments and your encouragements! We followed your comments to improve the revised MS, thanks again!
Title: The existing title does not accurately summarize the content of this study, and 'Enhance Rooting' is not comprehensive enough to cover the content, so it is suggested to change it to 'Enhance growth potential'.
Response: Many thanks for your comments!
The change in the title was done, thanks!
“Biological Nanofertilizers to Enhance Growth Potential of Strawberry Seedlings by Boosting Photosynthetic Pigments, Plant Enzymatic Antioxidants and Nutritional Status”
Figure 2-4: It is recommended to number the bar charts in the combination chart, such as A, B, C... or a, b, c....
Response: Many thanks for your comments!
The changes in all figures were done, thanks!
The position format of the horizontal legend 'Treatments' of the bar graph is inconsistent.
Response: Many thanks for your comments!
The changes in all figures were done, thanks! We changed them to be more consistent!
Discussion: Mentioned in 'Introduction': “the main objectives of this investigation were to: (3) investigate the optimal dose of the studied bio-nanofertilizers for the best nutritional status in strawberry seedlings under the conditions studied”; And with seven treatments (T1-T7) included in the experimental design, It is recommended that a comprehensive data analysis be conducted to conclude which treatment is more beneficial to the growth of strawberries and reflected in the abstract and conclusion.
Response: Many thanks for your comments!
More recommended issues were added to both abstract and conclusion
We added this section beside one more based on the comment of other review:
Nanofertilizers can support agro-food production by protecting cultivated plants and improving crop productivity [Guleria et al. 2023], for sustainable agriculture [Mahapatra et al. 2022], precision agriculture [Nongbet et al. 2022], improve food quality and safety [Yu et al. 2023], and of nano-farming [Haris et al. 2023].
The stability of nanoparticles/nanofertilizers is mainly controlled by the environmental conditions, which include rhizosphere and exudates of roots, soil pH, and salinity, soil CEC, soil mater content, soil pollution [Tiwari et al. 2022]. After forming synthetic NPs, they tend to evolve toward a more stable thermodynamic state, through interacting with surrounding molecules or undergoing physicochemical transformations such as corrosion, aggregation, and dissolution (Yu et al. 2023). NPs start linking with biomolecules (e.g., proteins, peptides, nucleic acids, lipids, and metabolites of cellular activities, and natural organic matter), which allow adsorbing NPs on surfaces immediately upon contact with a living system [Lv et al. 2022]. An expanding attention in the last decade was given to NPs-interactions within biological environments as well as their NPs interactions with ecological components (Yu et al. 2023).
Conclusions: It is better to avoid any general discussion in the conclusion part; please try to focus on the principal findings. It is suggested that the general discussion could be placed in 'Discussion'.
Response: Many thanks for your comments!
More improvements were added to the conclusion, thanks!
We added this part to the revised MS:
In general, applied nano-CuO at level of 100 ppm enhanced several studied parameters of strawberry seedlings such as some nutrient uptake (Cu, Mn, K, and Zn), survival rate of seedlings, and dry weight of seedlings. The remarkable increasing rate was recorded for Cu content in growing medium to be about 14-fold.

Round 2
Reviewer 2 Report
The Authors answered all the question and the MS has been improved well.